# DELI-FISHER GAN: STABLE AND EFFICIENT IMAGE GENERATION WITH STRUCTURED LATENT GENERATIVE SPACE

## ABSTRACT

Generative Adversarial Networks (GANs) are powerful tools for realistic image generation. However, a major drawback of GANs is that they are especially hard to train, often requiring large amounts of data and long training time. In this paper we propose the Deli-Fisher GAN, a GAN that generates photo-realistic images by enforcing structure on the latent generative space using similar approaches by DeliGAN as introduced in Gurumurthy et al. (2017). The structure of the latent space we consider in this paper is modeled as a mixture of Gaussians, whose parameters are learned in the training process. Furthermore, to improve stability and efficiency, we use the Fisher Integral Probability Metric as the divergence measure in our GAN model, instead of the Jensen-Shannon divergence. We show by experiments that the Deli-Fisher GAN performs better than DC-GAN, WGAN, and the Fisher GAN as measured by inception score.

## 1 INTRODUCTION

Generative Adversarial Networks (GAN) are powerful unsupervised learning models that have recently achieved great success in learning high-dimensional distributions(Goodfellow et al. (2014)). In the field of image and vision sciences in particular, GAN models are capable of generating "fake" images that look authentic to human observers.

The basic framework of a GAN model consists of two parts: a generator $G = G_\theta(z)$ that generates images by translating random input noise $z$ into a particular distribution of interest, and a discriminator $D = D_p(\mathbf{x})$ which calculates the probability that an image $\mathbf{x}$ is an authentic image as opposed to a generated "fake" image from the generator. While the generator $G$ and discriminator $D$ can be modeled as any smooth functions, these two components are usually modeled as two neural networks in practical applications. During the training process, we optimize the generator and the discriminator alternately against each other. Within each step, we first keep $D$ fixed and optimize $G$ so as to improve its capability of generating images that look real to $D$. Then, we keep $G$ fixed and train $D$ to improve the discriminator's ability to distinguish real and $G$-generated images. The two parts $G$ and $D$ play a two-player game against each other. At the end of the training, we would be able to have a generator that is capable of generating photo-realistic images.

In mathematical form, a GAN model can be described as an optimization problem, as follows:

$$\min_G \max_D V(D, G) \tag{1}$$

where $V(D, G)$ is the objective function measuring the divergence between the two distributions: the distribution of the real existing data $D(x)$, and the that of the generated data $D(G(z))$, where $x$ follows the distribution of real images and $z$ follows the distribution of input noise. Depending on the choice of function $V(D, G)$, different GAN models have been proposed over time (see Goodfellow et al. (2014), Arjovsky et al. (2017), Mroueh & Sercu (2017)) to increase stability and achieve faster convergence rates.

## 2 Previous work and problems

Ever since the inception of the first GAN models were introduced in Goodfellow et al. (2014), much improvement has been achieved on the GAN models. As mentioned in the previous section, the choice of the objective function $V(D, G)$ is crucial to the entire GAN model. The original GAN model in Goodfellow et al. (2014) optimizes the Jenson-Shannon divergence measure. This model, however, suffers from slow and unstable training. Some later work sought to improve GAN performance by utilizing the Earth-Mover Distance (Arjovsky et al. (2017)) and the more general f-divergences (Mroueh & Sercu (2017)), as well as other possibilities such as the Least Square Objective (Mroueh & Sercu (2017)). Along this line of research, one of the recent notable developments in GANs is the Fisher GAN model proposed by Mroueh & Sercu (2017), which employs the Fisher Integrated Probability Metric (Fisher IPM) to formulate the objective function.

In addition to the developments in divergences used as objective functions in GAN, recent research also focuses on the structure of the latent space for the generator. In particular, one of the 2017 CVPR papers Gurumurthy et al. (2017) introduced Deli-GAN, which uses input noise generated from the mixture of Gaussian distributions. The paper also argued that this method makes it possible to approximate a huge class of prior data distributions quickly by placing suitable emphasis on noise components, and hence makes training more efficient.

## 3 The Fisher IPM Framework

The loss function $V(D, G)$ as shown in (1) defines how we measure the difference between our learned distribution and the distribution from real images we want to learn. The divergence measure used in $V(D, G)$ directly controls what the model can achieve through the minimax optimization problem. Therefore, as shown by recent work, it is important to choose a stable and efficient divergence measure for the loss function. In the first GAN proposed in Goodfellow et al. (2014),the Jensen-Shannon divergence based on KL divergence between two distribution is used, but the model suffers from several problems such as unstable training and slow convergence. These inherent caveats prompted The WGAN proposed in Arjovsky et al. (2017) is more stable and only induces very weak topology (as weak as convergence in distribution), but is known to be costly in computation(Arjovsky et al. (2017),Gulrajani et al. (2017),Mroueh & Sercu (2017)).

In this paper, we choose to adopt the Fisher IPM framework proposed by Mroueh & Sercu (2017), which provides stability, efficient computation, and high representation power. Following the framework developed in Muller (1997), we define the Integral Probability Metric (IPM). Let $\mathcal{F}$ be the space of all measurable, symmetric, and bounded real functions. Let $\mathcal{X} \in \mathbb{R}^d$ be compact. Let $\mathbb{P}$ and $\mathbb{Q}$ be two probability measures on $\mathcal{X}$. Then the Integral Probability Metric is defined as

$$d_{\mathcal{F}}(\mathbb{P}, \mathbb{Q}) = \sup_{f \in \mathcal{F}} \left( \mathbb{E}_{x \sim \mathbb{P}} f(x) - \mathbb{E}_{x \sim \mathbb{Q}} f(x) \right)$$

Let $\mathcal{P}(\mathcal{X})$ denote the space of all probability measures on $\mathcal{X}$. Then $d_{\mathcal{F}}$ defines a pseudo-metric over $\mathcal{P}(\mathcal{X})$. By choosing an appropriate $\mathcal{F}$, we can define a meaningful distance between probability measures.

Now we define the Fisher IPM following the Fisher Discriminative Analysis framework as described in Mroueh & Sercu (2017). Given two probability measures $\mathbb{P}, \mathbb{Q} \in \mathcal{P}(\mathcal{X})$, the Fisher IPM between $\mathcal{P}$ and $\mathcal{Q}$ is defined as

$$d_{\mathcal{F}}(\mathbb{P}, \mathbb{Q}) = \sup_{f \in \mathcal{F}} \frac{\mathbb{E}_{x \sim \mathbb{P}} f(x) - \mathbb{E}_{x \sim \mathbb{Q}} f(x)}{\sqrt{\frac{1}{2} \mathbb{E}_{x \sim \mathbb{P}} f^2(x) + \frac{1}{2} \mathbb{E}_{x \sim \mathbb{Q}} f^2(x)}}$$

In order to formulate a loss function that is easily computable, we transform the above formula into a constrained format

$$d_{\mathcal{F}}(\mathbb{P}, \mathbb{Q}) = \sup_{f \in \mathcal{F}, \frac{1}{2} \mathbb{E}_{x \sim \mathbb{P}} f^2(x) + \frac{1}{2} \mathbb{E}_{x \sim \mathbb{Q}} f^2(x) = 1} \left( \mathbb{E}_{x \sim \mathbb{P}} f(x) - \mathbb{E}_{x \sim \mathbb{Q}} f(x) \right)$$

so that the problem is better suited for optimization, as we will see in the following sections.

## 4 DELI-FISHER GAN

Most GAN models introduced in previous work(Goodfellow et al. (2014),Arjovsky et al. (2017),Mroueh & Sercu (2017)) make use of random noise generated from a uniform distribution or a Gaussian distribution in their latent space for the input to the generator. These choices of using overly simplistic distributions, however, are not well justified. Since the data we train the GAN upon is often diverse with many varying classes of images. choosing one uniform or Gaussian distribution to generate the random noise input may fail to represent the features in the latent space.

We believe that a good choice of probability distribution for the latent noise will be able to translate into better features or structures in the generated image. An idea of using mixed Gaussian distribution in the latent space was proposed in Gurumurthy et al. (2017), in which the authors changed distribution of the random noise input from a singular uniform/Gaussian distributions to a mixture of Gaussians, and incorporated the GAN architecture from the DCGAN model described in Radford et al. (2015). During the training process, the parameters of the mixed Gaussian distribution (means and variances) are learned in each epoch. Once the training is complete, the Deli-GAN generates images using the mixed Gaussian learned from training process.

Thus, we incorporate this idea in our paper, and generalize the distribution of the latent space to general mixture distributions:

$$\mathcal{D}_{\text{latent}} = \sum_{i=1}^{N} w_i \mathcal{D}_{\theta_i}^{(i)}$$

where $\mathcal{D}_{\theta_i}^{(i)}$ are the distributions of the $i^{\text{th}}$ component, $\theta_i$ are the parameters of component, and $w_i$ are the weights for the component. For instance, if $\mathcal{D}_{\theta_i}^{(i)}$ are all Gaussian distributions, then $\theta_i = (\mu_i, \sigma_i)$ represent the means and standard deviations of these Gaussians. Using the mixture input random noise, we proceed to build the GAN model with the Fisher IPM we have described in the previous section.

The following sections will discuss in detail of the loss function and algorithms implemented.

### 4.1 THEORY AND THE LOSS FUNCTION

By our discussion above, we reformulate the Deli-Fisher GAN model into the following optimization problem:

$$\min_{g_\theta, \alpha_i, w_i} \sup_{f_p \in \mathcal{F}_p} V(f_p, g_\theta) := \mathbb{E}_{x \sim \mathbb{P}_r} f_p(x) - \mathbb{E}_{z_i \sim \mathbb{P}_g^{(i)}} f_p \left( g_\theta \left( \sum_i w_i z_{\alpha_i}^{(i)} \right) \right)$$

$$\text{subject to } \frac{1}{2} \mathbb{E}_{x \sim \mathbb{P}_r} f_p^2(x) + \frac{1}{2} \mathbb{E}_{z_i \sim \mathbb{P}_g^{(i)}} f_p^2 \left( g_\theta \left( \sum_i w_i z_{\alpha_i}^{(i)} \right) \right) = 1$$

(2)

where $\mathbb{P}_r$ is the distribution of the real images and $\mathbb{P}_g^{(i)}$ is the distribution of the $i^{\text{th}}$ component of latent input noise, as a multimodal distribution.

Empirical formulation of this loss function can vary depending on the distribution $\mathbb{P}_g^{(i)}$. In a simple case, if the $\mathbb{P}_g^{(i)}$'s are independently and identically distributed, and $\alpha_i$ only depend on their means $\mu_i$ and variances $\sigma_i$, i.e. $\alpha_i = \alpha_i(\mu_i, \sigma_i)$, then the empirical formulation of (2) can be written as

$$\min_{g_\theta, \alpha_i} \sup_{f_p \in \mathcal{F}_p} V(f_p, g_\theta) = \frac{1}{N} \sum_{i=1}^{N} f_p(x_i) - \frac{1}{M} \sum_{j=1}^{M} f_p(g_\theta(\mu_j + \sigma_j \epsilon_j))$$

$$+ \lambda(1 - \hat{\Omega}(f_p, g_\theta)) - \rho(\hat{\Omega}(f_p, g_\theta) - 1)^2 + \beta(C - \sigma)^2,$$

where

$$\hat{\Omega}(f_p, g_\theta) = \frac{1}{2N} \sum_{i=1}^{N} f_p^2(x_i) - \frac{1}{2M} \sum_{j=1}^{M} f_p^2(g_\theta(\mu_j + \sigma_j \epsilon_j)).$$

Here, $N$, $M$ are our sample sizes for the discriminator and the generator respectively, and $C$ is a constant controlling the size of $\sigma$. $\lambda$ represents the Lagrange multiplier for optimization, while $\rho$ and $\beta$ are penalty weights for the $L^2$-regularity of $\hat{\Omega}$ and $\sigma$, respectively. $\epsilon_i$ are random noises that provides diversity to the latent space. $\epsilon_i$ are sampled from the normalized $\mathbb{P}_g^{(i)}$. The parameters for our structured noise input are in turn updated during training process, as in the case with Gurumurthy et al. (2017).

## 4.2 DELI-FISHER GAN ALGORITHM

Using the standard stochastic gradient descent(SGD) algorithm ADAM, over all sets of parameters, we compute the updates of the respective variables by optimizing the loss functions described in the previous section with the following procedure:

---

**Input:** $\rho$ penalty weight, $\eta, \eta'$ learning rates, $n_c$ number of iterations for training the critic, $N$ batch size

Initialize $p, \theta, \lambda$
Initialize $\mu_i, \sigma_i, \eta$
**while** $\theta$ not converging **do**
    **for** $j = 1$ **to** $n_c$ **do**
        Sample a minibatch $x_i, i = 1...N, x_i \sim \mathbb{P}_r$
        Sample a minibatch $\epsilon_i, i = 1...N, \epsilon_i \sim$ normalized $\mathbb{P}_g^{(i)}$
        $(g_p, g_\lambda) \leftarrow (\nabla_p V, \nabla_\lambda V)(p, \theta, \lambda)$     // Using SGD with ADAM
        $p \leftarrow p + \eta g_p$
        $\lambda \leftarrow \lambda - \rho g_\lambda$
    **end**
    Sample $\epsilon_i \sim$ normalized $\mathbb{P}_g^{(i)}, i = 1, ..., M$
    $(d_\theta, d_\mu, d_\sigma) \leftarrow -\frac{1}{N} \nabla_{\theta,\mu,\sigma} \sum_{i=1}^N f_p(g_\theta(\mu_i + \sigma_i \epsilon_i))$
    $\theta \leftarrow \theta - \eta d_\theta$
    $\mu \leftarrow \mu - \eta' d_\theta$
    $\sigma \leftarrow \sigma - \eta' d_\theta$
**end**

**Algorithm 1:** Deli-Fisher GAN

---

# 5 EXPERIMENTS

## 5.1 QUALITY EVALUATION: INCEPTION SCORE

To evaluate the quality of the images generated by our GAN, we use the Inception Score as defined in Salimans et al. (2016), an automated measure aiming to simulate human judgment of quality of the images generated. This measure aims to minimize the entropy for the conditional label distribution $p(y|\mathbf{x})$ to ensure consistency between the generated images and given data, and maximize the entropy of the marginal $\int p(y|\mathbf{x} = G(z))dz$ to guarantee the diversity of the images generated by the network. In view of these two considerations, the proposed metric can be written as

$$\exp(\mathbb{E}_\mathbf{x}(D_{\mathrm{KL}}(p(y|\mathbf{x})\|p(y)))),$$

where $D_{\mathrm{KL}}(p\|q)$ denotes the Kullback-Leibler divergence between two distributions $p$ and $q$. An alternative measure involving the exponent of inception score has been proposed in Gurumurthy et al. (2017); for our experiments, we will stick to the original formulation as proposed in Salimans et al. (2016).

The inception score we used in all experiments below is calculated by the python script posted by OpenAI at `https://github.com/openai/improved-gan/tree/master/inception_score`.

## 5.2 DATASETS AND EXPERIMENTAL REPLICATIONS

As a baseline for subsequent comparison, we have replicated the experiments of previous GAN architectures. We have successfully replicated the results for Deep Convolutional Generative Adversarial Networks (DC-GAN) in Radford et al. (2015), Wasserstein GAN in Arjovsky et al. (2017), and Fisher GAN in Mroueh & Sercu (2017), all using the data set CIFAR-10. Table 1 and Table 2 are two tables that show the results of our experimental replication and the means and variances of their respective inception scores.

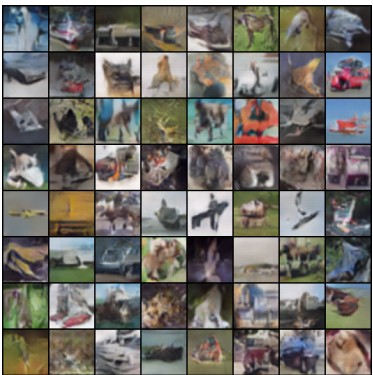

Figure 1: Sample generated images from DCGAN

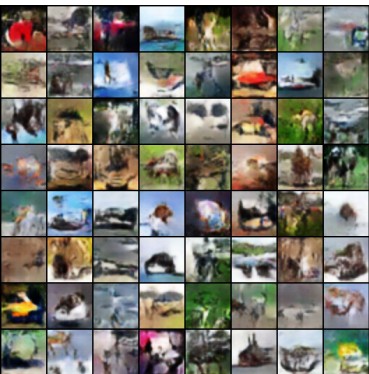

Figure 2: Sample generated images from WGAN

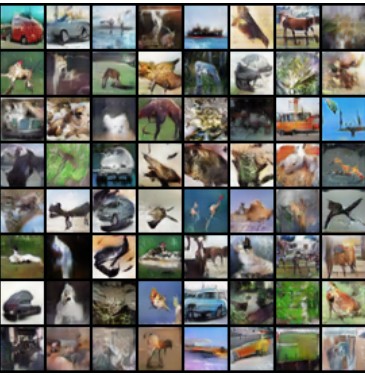

Figure 3: Sample generated images from FisherGAN

For the several experiments involving Deli-Fisher GAN, we used the CIFAR-10 dataset to examine our model. While all the images in the CIFAR-10 data are colored pictures with a size of $64 \times 64$,

| Type of GAN | Inception Score |
|---|---|
| DCGAN(as in Mroueh & Sercu (2017)) | (6.16, 0.07) |
| WGAN | (5.4586, 2.217e-2) |
| Fisher GAN | (6.8978, 0.09188) |

Table 1: Mean and Variance of Inception Scores in previous GANs

we used cropped images of size $32 \times 32$ so that the dense layer of our neural networks does not become too large. For each training, We generated 50,000 fake images and used these images to calculate the inception score.

Each the training session consists of 200 epochs. In each session, we applied generated corresponding output. Then we apply Deli-Fisher GAN to the same data set and compare the result with Fisher-GAN. In the Deli-Fisher GAN, we set hyper-parameters as 0 and initialized parameters for the input distribution ($\mu_i$, $\sigma_i$ and $\eta$). We executed same number of epochs in the training session. During the training session, $\theta, \mu, \sigma$ and $\eta$ were learned by Stochastic Gradient Descent with ADAM optimizer. After we have learned the parameters of the model, we generated another 50,000 images to make comparison with those generated by Fisher-GAN.

At the same time, we have also tuned different parameters in each model generation to fake sample production work-flow. These parameters include the number of epochs, the penalty coefficient, etc. We have also made use of the inception score described above to compare the images we've generated with the ones in the original data distribution.

All the experiments are done on GeForce GTX 1080Ti GPU, and we have observed that most of the GAN trainings involved in our experiments take around 30 minutes. One notable exception, however, lies in WGAN, since the weight-clipping procedures involved in WGAN requires a lot of computation and accounts for the extra time needed in experiments. Moreover, while repeating the experiments of different GANs, we noticed that the performances of DCGAN were highly unstable and unsatisfactory, as DCGAN yielded varying unsatisfactory inception scores at the range of 2 to 3 in our runs and stopped parameter updating even when the images are still blurred. These observations confirm the conclusions in Arjovsky et al. (2017) and Mroueh & Sercu (2017).

## 5.3 RESULTS

Using suitable parameters located through fine-tuning, the Deli-Fisher GAN produces better images than those produced by the current optimal Fisher GAN, as measured by the inception score. For comparison, the respective inception scores in experiments over the CIFAR-10 dataset are listed in Table 3.

| No. of Experiment | Inception Score |
|---|---|
| Experiment 1 | (7.3901662, 0.10778493) |
| Experiment 2 | (7.1032821, 0.055437498) |
| Experiment 3 | (7.4196534, 0.119866334) |
| Experiment 4 | (7.4921403, 0.0764598) |
| Experiment 5 | (7.2027034, 0.07927073) |

Table 2: Mean and variance of inceptions scores of Deli-Fisher GAN. Here the we used following set of parameters: learning rate $\sigma = 0.00005$. Latent space family: mixture Gaussian. In the table, each inception score has two values: the mean value and the standard deviation.

As demonstrated by the tables, experiments generated images of good qualities. One such sample is shown in Figure 4. Compared with previous GANs, we can see notable improvements in the images generated, by qualitative and quantitative observation. These outputs therefore suggest that a better representation of the random noise input does indeed capture more features of the latent space and those of the images the model is trained upon, and these features, in turn, augment the authenticity of the images that the Deli-Fisher GAN model produces.

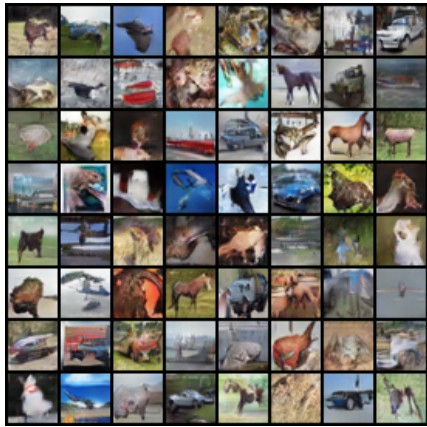

Figure 4: Sample generated images from Deli-Fisher GAN

## 6 CONCLUSION

In sum, the Deli-Fisher GAN presented in our paper is capable of generating better images than the DC-GAN, the WGAN, and the Fisher-GAN are, with notable improvements on the quality of images as measured by inception scores. Additionally, the model proposed in our paper is still open to improvement such as adding regularization terms to the objective function as those employed in the experiments of Mroueh & Sercu (2017).

As a further step, we are working on developing more sophisticated structures for the latent space that is specific tailored to different tasks. We believe, by enforcing some properties on the latent space, e.g. symmetries or geometric characteristics, we would be able to gain some control of the features on the generated images.

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
