# OpenReview forum: "Deli-Fisher GAN: Stable and Efficient Image Generation With Structured Latent Generative Space"
_ICLR.cc/2019/Conference_

### Official Review · AnonReviewer3 · 2018-11-05
**This paper presents a small modification to a previous GAN framework, and does not seem to be complete**

**Rating:** 3
**Confidence:** 5

**Review:**

- Summary
This paper presents a minor improvement over the previous Deli-GAN framework (Gurumurthy et al CVPR'17). Specifically, the work proposes to use the Fisher Integral Probability Metric as the divergence measure in our GAN model, instead of the Jensen-Shannon divergence. It shows little results (seems to be positive) using this new distance measure other than traditional ones.  Except that, I didn't see any other contribution from this paper.

- Suggestions
The paper is poorly written and seems to be a rush submission to ICLR. For example:
* a lot of grammatical errors throughout the paper
* only 6.5 papes out of 8 pages are utilized
* the introduction is not convincing -- what problem are you going to address? any summary of your methodology? why it is expected to outperform existing frameworks? what distinguishes your work from existing works? and what're your main results? I cannot conclude after reading the intro.
* the results are very minor and not convincing. It seems the authors conducted a very limited set of experiments and concluded that the proposed Deli-Fisher GAN is better. If you claim that the proposed framework can generate better images, at least the framework should be compared to the latest state-of-the-art GANs (e.g. spectral GANs, etc.)
* The writing is not polished.

Overall, the paper is far from ready to be submitted to ICLR, not mentioning acceptance. I would recommend the authors to conduct more experiments and comparisons and do a better job before submitting it to future conferences.

---

### Official Review · AnonReviewer2 · 2018-11-10
**Straightforward combination of prior works with no theory and weak experiments**

**Rating:** 2
**Confidence:** 5

**Review:**

This paper is a straightforward combination of two previous works, Deli-GAN and Fisher GAN. Deli-GAN has a mixture prior distribution in the latent space, while Fisher GAN uses Fisher IPM instead of JSD as objective.
Inception score on CIFAR-10 is used to empirically measure quality.

Cons:
The exposition of the ideas is lacking. What's wrong with Deli-GAN? What is this paper trying to accomplish by incorporating fisher metric?
No theoretical justification while empirical results are sparse and unconvincing.
Writing quality could be improved throughout the paper in terms of both structure and language.

In summary, this paper is not of the quality that should be accepted by ICLR.

---

### Official Review · AnonReviewer1 · 2018-11-12
**Trivial extension of previous paper; weak experiments.**

**Rating:** 2
**Confidence:** 4

**Review:**

The paper proposes to combine two ideas from previous publications, Fisher-GAN and Deli-GAN, i.e., use a mixture noise (Deli) with Fisher IPM metric for training GAN.

The extension of the previous work is trivial and the combination of the two ideas lack of any motivation. The experimental results are also weak. It is certainly below the bar of acceptance.

---

### Author Response · Authors · 2018-11-24
**Acknowledgement for reviewers and possible future plans**

Thank you so much for your pertinent reviews and suggestions.  We do realize the problems in writing(time is a bit rushed before submission) and will make effort to improve on experiments for submission to future venues.  We will also try to modify the structure of the paper and add comparisons to the existing models.

---

### Meta-Review · Area_Chair1 · 2018-12-05
**not ready for publication at ICLR**

**Confidence:** 5
**Recommendation:** Reject

**Metareview:**

This paper combines two recently proposed ideas for GAN training: Fisher integral probability metrics, and the Deli-GAN. As the reviewers have pointed out, the writing is somewhat haphazard, and it's hard to identify the key contributions, why the proposed method is expected to help, and so on. The experiments are rather minimal: a single experiment comparing Inception scores to previous models on CIFAR; Inception scores are not a great measure, and the experiments don't yield much insight into where the improvement comes from. No author response was given. I don't think this paper is ready for publication in ICLR.